

# Hard time to be parents? Sea urchin fishery shifts potential reproductive contribution of population onto the shoulders of the young adults

Barbara Loi[1,2], Ivan Guala[1], Rodrigo Pires da Silva[1], Gianni Brundu[1,2], Maura Baroli[1] and Simone Farina[1]

[1] IMC—International Marine Centre, Torregrande, Oristano, Italy
[2] Department of Ecological and Biological Sciences, University of Tuscia, Viterbo, Italy

Corresponding author
Barbara Loi, b.loi@fondazioneimc.it

## ABSTRACT

**Background.** In Sardinia, as in other regions of the Mediterranean Sea, sustainable fisheries of the sea urchin *Paracentrotus lividus* have become a necessity. At harvesting sites, the systematic removal of large individuals (diameter $\geq 50$ mm) seriously compromises the biological and ecological functions of sea urchin populations. Specifically, in this study, we compared the reproductive potential of the populations from Mediterranean coastal areas which have different levels of sea urchin fishing pressure. The areas were located at Su Pallosu Bay, where pressure is high and Tavolara-Punta Coda Cavallo, a marine protected area where sea urchin harvesting is low.

**Methods.** Reproductive potential was estimated by calculating the gonadosomatic index (GSI) from June 2013 to May 2014 both for individuals of commercial size (diameter without spines, TD $\geq 50$ mm) and the undersized ones with gonads ($30 \leq$ TD $< 40$ mm and $40 \leq$ TD $< 50$ mm). Gamete output was calculated for the commercial-size class and the undersized individuals with fertile gonads ($40 \leq$ TD $< 50$ mm) in relation to their natural density (gamete output per m$^2$).

**Results.** The reproductive potential of populations was slightly different at the beginning of the sampling period but it progressed at different rates with an early spring spawning event in the high-pressure zone and two gamete depositions in early and late spring in the low-pressure zone. For each fertile size class, GSI values changed significantly during the year of our study and between the two zones. Although the multiple spawning events determined a two-fold higher total gamete output of population (popTGO) in the low-pressure zone, the population mean gamete output (popMGO) was similar in the two zones. In the high-pressure zone, the commercial-sized individuals represented approximatively 5% of the population, with almost all the individuals smaller than 60 mm producing an amount of gametes nearly three times lower than the undersized ones. Conversely, the high density of the undersized individuals released a similar amount of gametes to the commercial-size class in the low-pressure zone.

**Discussion.** Overall, the lack of the commercial-size class in the high-pressure zone does not seem to be very alarming for the self-supporting capacity of the population, and the reproductive potential contribution seems to depend more on the total density of fertile sea urchins than on their size. However, since population survival in the high-pressure zone is supported by the high density of undersized sea urchins between 30 and 50 mm,

management measures should be addressed to maintain these sizes and to shed light on the source of the larval supply.

# INTRODUCTION

Commercial fisheries are one of the main causes of the deterioration of marine communities (*Jackson et al., 2001*). The systematic removal of fishery resources drastically reduces natural stocks and changes growth, production and recruitment of target species (*Pinnegar et al., 2000*). In extreme cases, overfished populations are still present in the community but no longer interact significantly with other species (*Estes, Duggins & Rathbun, 1989*). This phenomenon can lead to a simplification of trophic webs with dramatic consequences for marine ecosystems (*Pauly, 1995*; *Myers & Worm, 2003*; *Coll, Lotze & Romanuk, 2008*; *Coll, Palomera & Tudela, 2009*; *Coll et al., 2009*; *Lotze, Coll & Dunne, 2011*; *Navia et al., 2012*; *Tunca et al., 2016*).

One of the clearest examples reported for several temperate coastal systems is the overexploitation of target species involved in the typical tri-trophic interaction "fish-sea urchins-macrophyte." Trophic relationships can be altered by overfishing predatory fishes. This triggers an uncontrolled proliferation of sea urchins which then leads to an overgrazing of algal cover (*Sala, Boudouresque & Harmelin-Vivien, 1998*; *Steneck et al., 2002*; *Steneck, Vavrinec & Leland, 2004*). Although this is the main effect of overfishing on many Mediterranean coasts (*Sala & Zabala, 1996*; *Sala, Boudouresque & Harmelin-Vivien, 1998*), trophic imbalances have also been found to act in the opposite direction. Indeed, over the last 2–3 decades, the general decline of natural fish stocks has led to a focus on new target species, often further down the food web.

In many regions sea urchin harvesting was added to higher-trophic-level fisheries (*Anderson et al., 2011*). This is the case of some regions in the South of Italy where the edible sea urchin *Paracentrotus lividus* (Lamarck, 1816) is subjected to high fishing pressure (*Tortonese, 1965*; *Guidetti, Terlizzi & Boero, 2004*; *Pais et al., 2007*). The most striking effect of sea urchin fishing is the rapid decrease in commercial resources in terms of total density and abundance (*Andrew et al., 2002*; *Bertocci et al., 2014*). A remarkable reduction in density and mean size of the sea urchin population would have dramatic consequences for the whole benthic community. For instance, the removal of hundreds of thousands of sea urchins from the temperate reef has coincided with the rapid development of brown algae that has led to substantial changes in the abundance of fish and benthic invertebrates (*Bell et al., 2014*).

Furthermore, the decline of sea urchin populations due to harvesting by humans could be even more significant because of the loss of sexually mature individuals that contribute to the local recruitment pool (*Levitan & Sewell, 1998*). Indeed, gonads are proportional to a sea urchin's body-size and are more mature and developed in large sea urchins (*Mita et al., 2007*).

The gonadosomatic index (GSI) is generally used to evaluate the reproductive features of echinoids such as fluctuations in gonad size and spawning periods (*Spirlet, Grosjean & Jangoux, 1998*; *Shpigel et al., 2004*; *Gianguzza et al., 2013*) and these relate to the reproductive potential of the individual (*Brewin et al., 2000*). Fertile size classes can produce more than one cohort of mature gametes in a single breeding season (*Mita et al., 2007*) and the reproductive cycle generally has one or two seasonal GSI peaks (see reviews in *Boudouresque & Verlaque, 2007* and *Ouréns, Fernández & Freire, 2011*). Sometimes there can be continuous spawning events of lower significance and with strong dependence on the variability of the gametogenesis (*Boudouresque & Verlaque, 2007*). Generally, GSI shows considerable spatial and temporal variability as a result of extrinsic factors such seasonal changes in photoperiod, water temperature and phytoplankton blooms (see *Ouréns, Fernández & Freire, 2011* and references therein). Food quality and availability (*Byrne, 1990*; *Minor & Scheibling, 1997*; *Brady & Scheibling, 2006*; *Scheibling & Hatcher, 2007*) as well as hydrodynamic conditions (*Lozano et al., 1995*; *Meidel & Scheibling, 1998*; *Guettaf, San Martin & Francour, 2000*; *Sellem & Guillou, 2007*; *Gianguzza et al., 2013*) can also influence the reproductive cycle and fecundity of populations.

In Sardinia (Italy, Western Mediterranean), commercial fishing of the sea urchin *Paracentrotus lividus* is limited by law to specimens larger than 50 mm test diameter (TD), from November to April. However, despite regional decrees concerning fishing periods, minimum size and catch quotas per day per fisherman, the harvesting of *P. lividus* is intensively practiced. Removal by occasional recreational fishermen occurs throughout the year because of the long tourist season (*Pais et al., 2007*). The systematic removal of the largest sea urchins may decrease the number of fertile individuals that release gametes into the surrounding environment. This leads to a population collapse, as reported for some overfished areas (*Pais et al., 2007*).

The aim of this work is to assess the reproductive potential of populations living in two zones of Sardinia which have different harvesting pressure and to compare, through the evaluation of the GSI over one year, the annual gamete output (*Brewin et al., 2000*) of the commercial-size class (diameter without spines, TD $\geq$ 50 mm) and the fertile, undersized one ($40 \leq TD < 50$ mm) of both populations. Our hypothesis supports that, under high fishing pressure, the contribution of the commercial-size class to reproductive potential drastically decreases and, as a result, the risk of a population collapse increases.

## MATERIALS AND METHODS

### Study sites and sea urchin sampling

The reproductive potential of the *Paracentrotus lividus* populations was examined in two zones of Sardinia that differ in sea urchin fishing pressure (Fig. 1). These two zones were selected because they are at two extremes as regards the exploitation of sea urchins in Sardinia. Su Pallosu Bay, located along the Sinis peninsula (central-western Sardinia, 40°03′N; 008°25′E), is subjected to very high pressure (HP zone) which is widespread across the entire bay. Harvesting can be practised through scuba diving from November to April by 189 professional fishermen authorized by regional decree. Each one of them, when
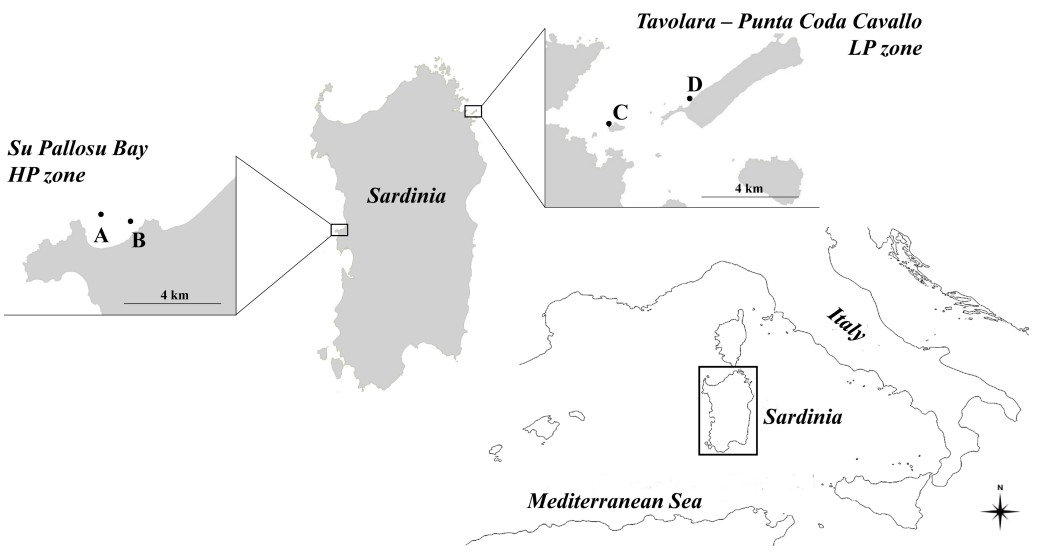

**Figure 1 Map of the study zones.** Su Pallosu Bay, in the Sinis Peninsula, is considered a high-pressure zone (HP zone); Tavolara—Punta Coda Cavallo is considered a low pressure zone (LP zone). Sampling areas, named (A) and (B) for Su Pallosu Bay, (C) and (D) for Tavolara—Punta Coda Cavallo, are represented by the black dots.

helped by an assistant, is allowed to collect up to 3,000 sea urchins (TD $\geq$ 50 mm) per day, according to the regional decree n. 1967/DecA/67 issued by the Regione Autonoma Sardegna (*RAS, Regione Autonoma della Sardegna 2013*).

On the contrary, the marine protected area of Tavolara—Punta Coda Cavallo (northeastern Sardinia, 40°53′N; 009°40′E) is considered a low-pressure zone (LP zone). This marine reserve was legally established in 1997 and has been considered well-enforced since 2003 (*Di Franco et al., 2009*). This is one of the few zones in Sardinia where sea urchin fishing is strongly restricted (there are only few authorized fishermen and no more than 500 catches per day for an overall maximum of 300,000 sea urchins harvested each year) and the populations are well preserved (*Guala et al., 2011*). Although no data are available on annual catches, Su Pallosu Bay is among the most exploited in the whole region and the theoretical number of sea urchins that can be legally harvested in one year (calculated according to the number of licenses, allowed catches and fishing days) is over 300 times greater than that of Tavolara—Punta Coda Cavallo.

Sea urchin specimens were collected from two areas of high harvesting pressure (A and B) at Su Pallosu Bay, and low harvesting pressure (C and D) at Tavolara MPA. In each zone, the two areas were 1–3 km apart. Sampling (approved by the Regione Autonoma Sardegna through the release of the fishing license for scientific purposes n. 9727/AP SCIE/N.7 03/06/2016) involved all sea urchin size classes that have gonads and contribute to the reproductive potential of the population. Specifically, once a month, we collected 10 individuals of the commercial size (CS, TD $\geq$ 50 mm), five undersized individuals (US, 40 $\leq$ TD < 50 mm) and five smaller, undersized sea urchins (Small-US, 30 $\leq$ TD < 40 mm) for an entire year.
The sea urchins were collected from both zones by scuba diving over a rocky bottom covered in photophilic algal communities at a depth of five meters. Sea urchin samples were wrapped in cloth soaked in salt water, stocked in iceboxes and immediately transported to the laboratory.

## Environmental features

Despite the differences in position along the Sardinian coast and in wind exposure, sampling was performed in areas with similar environmental features (i.e., depth, slope, shelter from the waves), on rocky bottoms.

Differences in food availability, potentially able to affect sea urchin gonad growth and consequently fecundity (*Byrne, 1990*; *Minor & Scheibling, 1997*; *Brady & Scheibling, 2006*; *Scheibling & Hatcher, 2007*), were assessed as temporal changes of algal cover between zones. Digital photographs were taken over three PVC quadrats of 50 × 50 cm which were randomly placed on the sea bottom in each area. This was done three times (July 2013, January and May 2014) during the surveyed year. Image analysis was carried out using Seascape® software (Segmentation and Cover Classification Analyses of Seabed Images, *Teixidó et al., 2011*) to detect the percentage cover of conspicuous algal taxa or morphological groups. A permutational multivariate analysis of variance (PERMANOVA, *Anderson, 2001a*) was done, on the basis of a Bray–Curtis dissimilarity matrix calculated from square-root transformed data (Primer-E 6 Permanova®), to estimate the variability of assemblages between the two zones over time. A 3-way model was used with Time (random, 3 levels) and Zone (fixed, 2 levels, Su Pallosu Bay vs. Tavolara—Punta Coda Cavallo) as crossed factors, and Area (2 levels) as a random factor nested in Zone. *P*-values were obtained through Monte Carlo random draws from the asymptotic permutation distribution (*Anderson, 2005*) and a pairwise test was used to discriminate between various levels of significant factors. A non-metric Multi-Dimensional Scaling (nMDS) ordination was used as a graphical representation of data.

In order to investigate any potential thermic anomalies, seasonal variations of the coastal water temperature were monitored with the "Mediterranean Sea—High Resolution and Ultra High Resolution L3S Sea Surface Temperature" product (http://marine.copernicus.eu/web/69-myocean-interactive-catalogue.php). Daily sea surface temperatures (SST) were extrapolated from the catalogue after choosing an intermediate point between the two sampling areas within the zones, then mean monthly temperatures were obtained and used to represent the annual trend.

## Gonadosomatic Index and fertility

The gonadosomatic index (GSI) was examined every month from June 2013 to May 2014 (with the exception of November due to adverse weather conditions) for all of the three sampled size classes. Sea urchins were allowed to drip dry for some minutes and then weighed. The test diameter (TD) without spines was measured and the gonads were successively extracted and weighed as well.

GSI was calculated by the formula: GSI = [gonad wet weight/total wet weight] × 100 as reported by *Lawrence, Lawrence & Holland (1965)*.

The fertility was tested according to *Brundu et al. (2016)* during the months of maximum gonadal development (i.e., from December to April). Gonads from females were extracted and gently shaken in filtered seawater to allow the mature ova to come out. Sperm was then added and finally, the fertility of the individuals was assessed as a percentage of the effectively fertilized eggs after the appearance of the fertilization membrane (fertilization was considered successful if it took place in at least 80% of the eggs according to *Falugi & Angelini, 2002*). The percentage of achievement of the first larval stage (development of four-arm echinopluteus) was also measured. The fertility test was carried out on the US individuals but not on Small-US specimens because of the paucity of their gonads (see "Results"). Nor was it done on the commercial-sized individuals (CS) because they were assumed to be fertile (*Ouréns, Fernández & Freire, 2011*).

Monthly mean values of GSI were calculated for the two zones and areas of sampling. Since sex ratio was nearly 1:1 for both sampling zones (Data S8), females and males were pooled together to obtain a single mean GSI value per month. A four-way ANOVA (using Statistica 6.0, Statsoft Inc.) was performed to highlight the differences in GSI values in different months (fixed and orthogonal factor, 11 levels), zones (fixed and orthogonal factor, 2 levels), areas (random and nested in zones, 2 levels), and fertile size classes (fixed and orthogonal factor, 2 levels, CS and US). Eight replicates for each size class were haphazardly selected from among those available in order to get a balanced design. Cochran's $C$ test was used to check for the assumption of homogeneity of variances and a posteriori SNK tests were performed to find alternative hypotheses (*Underwood, 1997*).

## Population structure and potential reproductive contribution

Abundance and size-frequency distribution of the populations were estimated for both zones by counting all sea urchins found in the PVC plots and measuring them with calipers. The plots were 50 × 50 cm and were placed randomly as many times as necessary to cover two replicates of 25 m$^2$ in each area (100 m$^2$ for each zone).

Sea urchin abundance was estimated as the total density (individuals m$^{-2}$) and the density of each 10 mm size class: TD < 10 mm, $10 \leq$ TD < 20 mm, $20 \leq$ TD < 30 mm, $30 \leq$ TD < 40 mm, $40 \leq$ TD < 50 mm and $50 \leq$ TD < 60 mm, TD $\geq$ 60 mm. Size-class densities were then translated into frequency percentages and used to compose the population structure for each zone. The gamete output and the spawning magnitude of each spawning event were calculated for the fertile CS and US classes according to *Brewin et al. (2000)*. The highest and the lowest mean monthly GSI recorded during the year of sampling corresponded to the period just before the beginning (pre-spawning) and after the end (post-spawning) of the spawning events. We defined the mean individual gamete output (IGO), in units of gamete wet weight per urchin per spawning event (g g$^{-1}$ se$^{-1}$), as the difference of the mean monthly pre-spawning GSI and the mean monthly post-spawning GSI. The spawning magnitude was defined as the percentage ratio of the mean individual gamete output and the mean monthly pre-spawning GSI.

For each spawning event, we calculated the gamete output per m$^2$ (GO, g g$^{-1}$ m$^{-2}$ se$^{-1}$), released by fertile size classes in both zones, as their IGO multiplied for the respective natural density. The total gamete output (TGO) and the mean gamete output (MGO), used to

estimate the reproductive contribution of each size class per m$^2$ per year, were defined respectively as the sum and the average of GO (g g$^{-1}$ m$^{-2}$ yr$^{-1}$). Finally, the potential reproductive contribution of the whole population in both zones (popTGO and popMGO, respectively) were calculated for the investigated year as the sum of the contributions of both fertile size classes.

# RESULTS

## Environmental features

Multivariate analysis on algal cover has detected a significant interaction between Time and Zone. More specifically, pairwise tests highlighted that algae where sea urchins live changed significantly over time but not between the two zones (Table S1). The nMDS showed that plots of the HP zone were interspersed on the graph with those of the LP zone (Fig. S2).

Moreover, the water temperature changed over time but no difference was observed in the annual trend between the two zones studied (Fig. S3).

## Gonadosomatic Index and fertility

At each area, 220 specimens of the commercial-size class (CS) and 110 undersized and smaller undersized individuals (US and Small-US) were randomly collected over a year to compare reproductive potential between populations. Regarding CS individuals, sampled sea urchins with TD $\geq$ 60 mm were 24 of 440 (5.5%) in the HP zone and 219 of 417 (53%) in the LP zone.

Fertility tests showed that 100% of the undersized individuals (US) checked were fertile and contributed to the reproductive potential of the populations, with high percentages of fertilized eggs (ranging from 87 to 96%) and developing larvae (ranging from 95 to 100%) (Table S4). Conversely, the Small-US individuals had reduced gonads (Fig. 2) and, even during the period of maximum development, they never released gametes, therefore their contribution to the reproductive potential of the population can be considered negligible.

The GSI trend over the year was generally higher for CS individuals than US ones for both zones (Fig. 2). At the HP zone, we recorded a single large spawning period from March to May for both fertile classes (see Fig. 2A). In pre-spawning time, GSI values reached 6.6 $\pm$ 0.3% (mean $\pm$ standard error) and 4.4 $\pm$ 0.4%, while in post-spawning time values were 1.3 $\pm$ 0.2% and 1.6 $\pm$ 0.1% for the CS and US classes respectively (Fig. 2A).

At the LP zone, a spawning event was observed twice (Fig. 2B). The first one was recorded from June to December with a pre-spawning GSI of 6.7 $\pm$ 0.3% and 5.2 $\pm$ 0.5%, and a post-spawning GSI of 2.5 $\pm$ 0.3% and 1 $\pm$ 0.2% for the CS and US classes respectively. The second spawning event occurred from February to April with a pre-spawning GSI of 5.4 $\pm$ 0.4% and a post-spawning GSI of 2.5 $\pm$ 0.2% for the CS class. From February to May we observed a pre-spawning GSI of 4 $\pm$ 0.6% and a post-spawning GSI of 1.2 $\pm$ 0.2% for the US class (Fig. 2B).

ANOVA highlighted significant differences during the year and between the different size classes, while there were no major statistical differences found between sampling areas and zones. The CS class had higher GSI values in the LP zone while no differences were found for the US individuals (Table 1). A significant interaction between Zone and Size

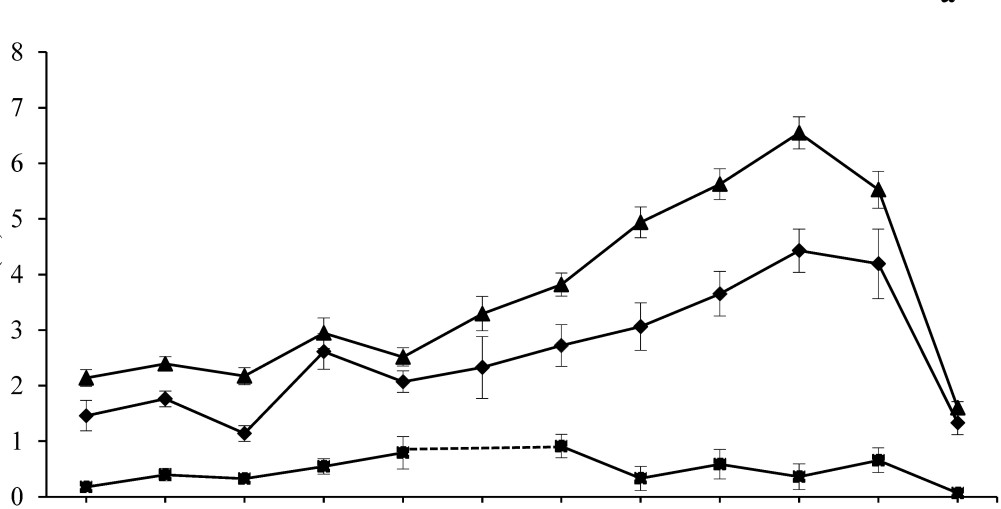

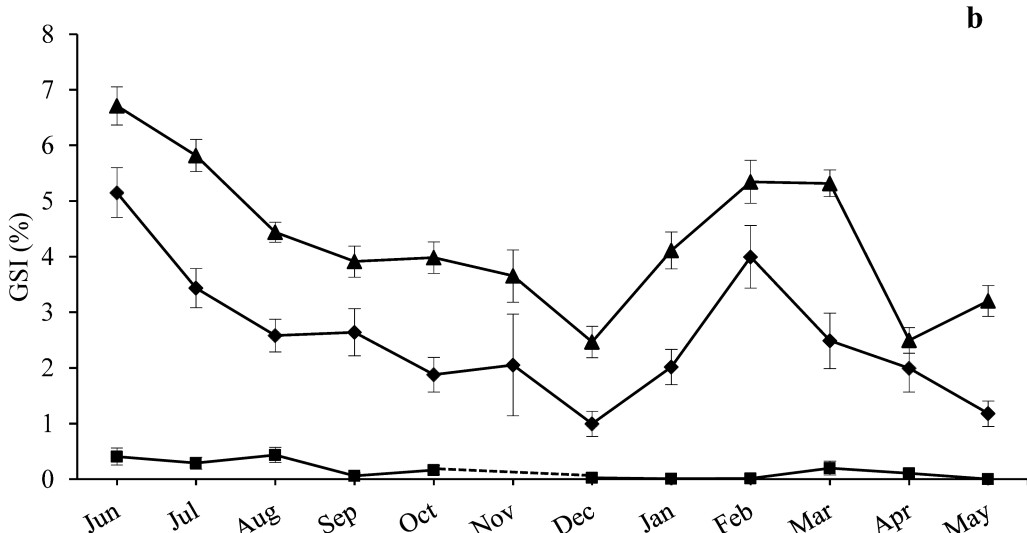

**Figure 2 Annual trend of gonadosomatic index at sampling zones.** GSI is represented as mean ± standard error for the three size classes examined (CS, US, Small-US) at (A) high-pressure zone (Su Pallosu Bay) and (B) low-pressure zone (Tavolara—Punta Coda Cavallo). No distinction between sexes was done in the graph, since females and males belonging to the same size class were pooled together. Observation began in June 2013 and ended in May 2014. However, November 2013 was not sampled due to general bad weather conditions and GSI data of CS and US classes were estimated from different years (i.e., November 2007 for Tavolara—Punta Coda Cavallo and November 2015/16 for Su Pallosu Bay). GSI data for Small-US class was not available for this month; the dotted line for the Small-US class represents an approximation of the expected values of GSI for November. Triangles, CS; rhombuses, US; squares, Small-US.

**Table 1 Results of four-way ANOVA.** Analysis of variance was performed to test the effects on gonado-somatic index of Month, Zone and Size class (orthogonal fixed factors) and Area (random nested factor in Zone). Bold value is statistically significant at $p < 0.05$. SNK tests were conducted for comparisons of significant interactions.

| Source of variation | df | MS | F | p |
|---|---|---|---|---|
| **Month = Mo** | 10 | 933.74 | 2.94 | **0.0193** |
| Zone = Zo | 1 | 11.883.74 | 14.26 | 0.0635 |
| Area(Zone) = Ar(Zo) | 2 | 833.46 | 2.27 | 0.1039 |
| **Size class = Sc** | 1 | 117763.64 | 362.25 | **0.0027** |
| Mo × Zo | 10 | 597.33 | 1.88 | 0.1104 |
| Mo × Ar(Zo) | 20 | 317.95 | 0.87 | 0.6301 |
| **Mo × Sc** | 10 | 2000.32 | 7.12 | **0.0001** |
| **Zo × Sc** | 1 | 13457.07 | 41.39 | **0.0233** |
| Sc × Ar(Zo) | 2 | 325.10 | 0.89 | 0.4126 |
| Mo × Zo × Sc | 10 | 648.71 | 2.31 | 0.0535 |
| Sc × Mo × Ar(Zo) | 20 | 281.10 | 0.77 | 0.7552 |
| Residual | 616 | 366.70 | | |

**Notes.**

Transformation: none; Cochran's test $C = 0.0664$, $p < 0.05$.
SNK Zo × Sc: High-pressure zone (US < CS); Low-pressure zone (US < CS); US (High-pressure zone = Low-pressure zone); CS (High-pressure zone < Low-pressure zone).
Mo × Sc: April (US = CS); Other months (US < CS).

Class was detected, and the GSI of the CS class was significantly higher than that of the US individuals for both zones. A significant interaction was also found between Month and Size Class with SNK pointing out significantly higher GSI values for the CS individuals during the whole sampling year excluding April (Table 1).

## Population structure and potential reproductive contribution

Size class distribution was consistently different between the two zones (Fig. 3). In the HP zone, sea urchin density was almost two-fold higher than in the LP zone: $10 \pm 1.4$ and $5.4 \pm 0.5$ individuals per $m^2$ respectively. Sea urchins with TD ranging from 0 to 20 mm were $1.7 \pm 0.1$ per $m^2$ and $1.1 \pm 0.6$ per $m^2$ in the HP and LP zones respectively, and they represent 17% and 21% of their populations. The most abundant size classes were those ranging from 20 to 50 mm diameter (77%) with a density of $3 \pm 0.6$ and $3 \pm 0.3$ individuals per $m^2$ for the Small-US and US classes respectively ($30 \leq TD < 40$ mm and $40 \leq TD < 50$ mm). The proportion of individuals of the CS class with respect to the entire population was 6% ($0.6 \pm 0.2$ individuals per $m^2$) and all the individuals were included in the range of $50 \leq TD < 60$ mm (Fig. 3). In the LP zone, the individuals between 20 and 50 mm represented 28% of the population while the CS class was 52% with $1 \pm 0.4$ and $2 \pm 0.2$ individuals per $m^2$ for $50 \leq TD < 60$ mm and $TD \geq 60$ mm respectively (Fig. 3).

In relation to the population structure, the reproductive contribution was compared between the two zones, but no comparisons were made between the sampling areas because no differences were found (see Table 1). The potential reproductive contribution was calculated according to the number of spawning events during the surveyed year and the natural density of the fertile size classes (US and CS). In the HP zone, a single spawning

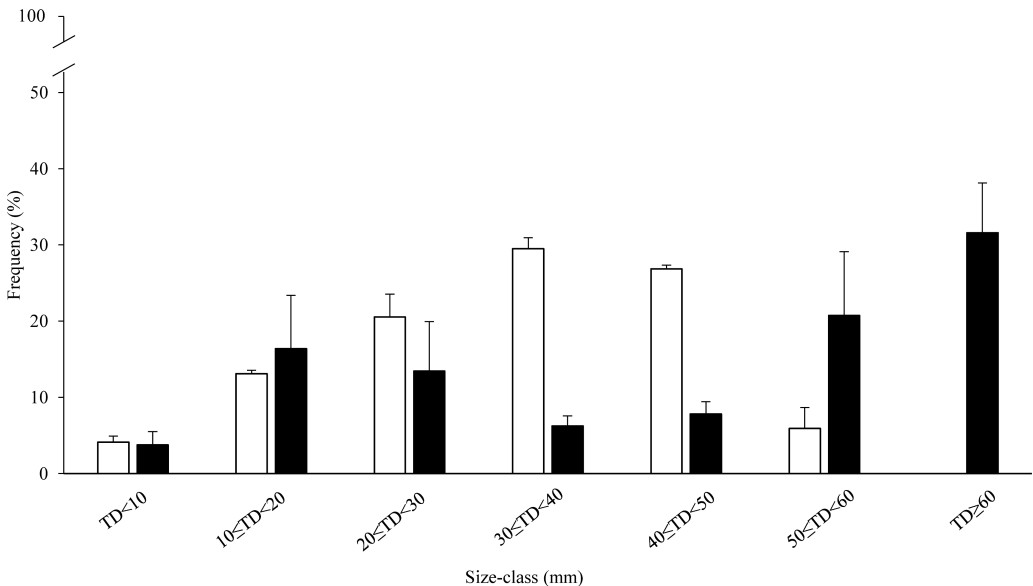

**Figure 3** **Size-frequency distribution (%) of sea urchin populations at sampling zones.** The range of the size classes is 10 mm of test diameter without spines (TD). Commercial size classes under fishing pressure are those larger than 50 mm test diameter. White bars, high-pressure zone (Su Pallosu Bay); black bars, low-pressure zone (Tavolara—Punta Coda Cavallo).

event occurred and the spawning magnitude for the year was 73% on average (Table 2) with an individual gamete output (IGO) of 0.03 and 0.05 g g$^{-1}$ se$^{-1}$ for the US and CS individuals, respectively (Table 2). The gamete output (GO), calculated in relation to the natural density of this zone, was 0.08 g g$^{-1}$ m$^{-2}$ se$^{-1}$ for the US sea urchins (2.7 $\pm$ 0.3 individuals per m$^2$) and 0.03 g g$^{-1}$ m$^{-2}$ se$^{-1}$ for the CS class (0.6 $\pm$ 0.2 individuals per m$^2$). Because of the single spawning event, the total gamete output per m$^2$ (TGO) overlaps the mean gamete output per m $^2$ (MGO) (Table 2). Accordingly, the total gamete output of the whole population (i.e., sum of TGO of the two fertile size classes) corresponded to the mean gamete output for m$^2$ (popMGO) with a value of 0.11 g g$^{-1}$ m$^{-2}$ yr$^{-1}$ (Table 2).

Conversely, in the LP zone, two spawning events were observed (Fig. 2B). Spawning magnitude varied from 54 to 81% with higher values for US individuals (Table 2). IGO was similar for both size classes with values ranging from 0.03 to 0.04 g g$^{-1}$ se$^{-1}$ according to the spawning period. The GO of the US individuals was 0.02 g g$^{-1}$ m$^{-2}$ se$^{-1}$ during the first spawning event and 0.01 g g$^{-1}$ m$^{-2}$ se$^{-1}$ during the second one, while it was 0.11 and 0.08 g g$^{-1}$ m$^{-2}$ se$^{-1}$ for the CS class. Total gamete output (TGO) and mean gamete output (MGO) of the US individuals, whose density was 0.4 $\pm$ 0.1 individuals per m$^2$, were 0.03 and 0.01 g g$^{-1}$ m$^{-2}$ yr$^{-1}$ respectively. Meanwhile they were 0.19 and 0.10 g g$^{-1}$ m$^{-2}$ yr$^{-1}$ for the CS class, whose natural density was 2.7 $\pm$ 0.3 individuals per m$^2$. Consequently, the total gamete output of the whole population was estimated to be 0.22 g g$^{-1}$ m$^{-2}$ yr$^{-1}$ and the total mean gamete output was 0.11 g g$^{-1}$ m$^{-2}$ yr$^{-1}$ (Table 2).

**Table 2  Summary table of potential reproductive contribution results.** Timing and extent of mean monthly gonadosomatic index, spawning magnitude, and gamete output for US ($40 \leq$ TD $< 50$ mm) and CS (TD $\geq 50$ mm) class, for spawning event and in relation with their natural density, and for the whole populations at high-pressure (Su Pallosu Bay) and low-pressure (Tavolara—Punta Coda Cavallo) zone.

| | | High-pressure zone | | Low-pressure zone | | | |
|---|---|---|---|---|---|---|---|
| Size class | | US | CS | US | | CS | |
| Spawning period | Start | Mar '14 | Mar '14 | Jun '13 | Feb '14 | Jun '13 | Feb '14 |
| | End | May '14 | May '14 | Dec '13 | May '14 | Dec '13 | Apr '14 |
| Mean monthly GSI (%) | Pre-spawning | 4.4 | 6.6 | 5.2 | 4.0 | 6.7 | 5.4 |
| | Post-spawning | 1.3 | 1.6 | 1.0 | 1.2 | 2.5 | 2.5 |
| IGO (g g$^{-1}$ se$^{-1}$) | | 0.03 | 0.05 | 0.04 | 0.03 | 0.04 | 0.03 |
| Spawning magnitude (%) | | 70.5 | 75.8 | 80.8 | 70.0 | 62.7 | 53.7 |
| Natural density (ind m$^{-2}$) | | 2.7 | 0.6 | 0.4 | | 2.7 | |
| GO (g g$^{-1}$ m$^{-2}$ se$^{-1}$) | | 0.08 | 0.03 | 0.02 | 0.01 | 0.11 | 0.08 |
| TGO (g g$^{-1}$ m$^{-2}$ yr$^{-1}$) | | 0.08 | 0.03 | 0.03 | | 0.19 | |
| popTGO (g g$^{-1}$ m$^{-2}$ yr$^{-1}$) | | 0.11 | | | 0.22 | | |
| MGO (g g$^{-1}$ m$^{-2}$ yr$^{-1}$) | | 0.08 | 0.03 | 0.01 | | 0.10 | |
| popMGO (g g$^{-1}$ m$^{-2}$ yr$^{-1}$) | | 0.11 | | | 0.11 | | |

**Notes.**
IGO, Individual Gamete Output per spawning event; GO, Gamete Output per m$^2$; TGO, Total Gamete Output per m$^2$; popTGO, Total Gamete Output of the whole population per m$^2$; MGO, Mean Gamete Output per m$^2$; popMGO, Mean Gamete Output of the whole population per m$^2$.

## DISCUSSION

The results showed a considerable difference between the two zones in population structure and in abundance of sea urchins, as well as in the gonadosomatic index trend throughout the surveyed year. As a consequence, even the potential reproductive contribution differed between the two populations.

While recruits (TD $< 20$ mm) had the same proportion of population structure, the density of the fertile undersized individuals (US) was 4.5-fold higher at the HP zone than at the LP one, and the density of the commercial size (CS) was $\sim$7-fold lower. Furthermore, within the commercial size, a large percentage was composed of individuals $\geq 60$ mm at the LP zone, while they were nearly absent at the HP zone (none have been detected during the sampling for the estimation of the population structure, only 24 throughout the annual sampling for the GSI assessment). Both populations showed a spawning event at the end of winter or at the beginning of spring, which lasted until April–May for the two fertile size classes examined (US and CS). Moreover, a second relevant peak of GSI was registered in the LP zone which was demonstrated to be concomitant with a spawning event from June 2013 (*Siliani et al., 2016*) by histological analysis. This is consistent with observations from other areas of the Mediterranean where one or two annual spawning periods were commonly identified, regardless of the proportion of size classes shaping the population structure (*Fenaux, 1968*; *Semroud & Kada, 1987*; *Pedrotti & Fenaux, 1992*; *Pedrotti, 1993*; *Semroud, 1993*; *Lozano et al., 1995*; *Fernandez & Boudouresque, 1997*; *López et al., 1998*; *Guettaf, San Martin & Francour, 2000*; *Leoni et al., 2003*; *Martínez et al., 2003*; *Sánchez-España, Martínez-Pita & García, 2004*; *Tomas, Romero & Turon, 2004*; *Sellem & Guillou, 2007*).

The differences identified in reproductive potential between the populations were not ascribed to different environmental features since the sampling zones were similar in terms of sheltering from waves and there were no differences found for either water temperature or algal assemblages. Therefore, since fecundity depends on food availability and other environmental characteristics (*Minor & Scheibling, 1997*; *Scheibling & Hatcher, 2007*), the two populations could be considered similar in their ability to produce offspring.

At both zones, GSI of the Small US individuals was lower than 1% throughout the year, therefore its contribution to the reproductive potential of the population was considered negligible. On the contrary, GSI of the two fertile size classes (US and CS) changed significantly over the year and between zones. Specifically, GSI of the CS class was significantly higher than those of the US individuals and it was significantly higher in the LP zone than in the HP zone.

Growth rate is one of the main factors that may influence the size-frequency distribution of a sea urchin population (*Dix, 1972*; *Barry & Tegner, 1990*), and it can differ among populations (e.g., *Sellem, Langar & Pesando, 2000*; *Turon et al., 1995*). However, a previous study conducted in the same zones by *Loi et al. (2013)* showed that the growth rates of the two populations were comparable (Fig. S5). Definitively, as widely described in many other locations of the Mediterranean (*Guidetti, Terlizzi & Boero, 2004*; *Gianguzza et al., 2006*; *Pais et al., 2007*; *Ceccherelli, Pinna & Sechi, 2009*; *Bertocci et al., 2014*), our study clearly points out how human predation adversely affects the population structure in the HP zone, truncating the adult cohorts with TD ≥ 50 mm. The high presence of commercial-size adults (TD ≥ 50 mm) at the LP zone, and in particular the high number of individuals larger than 60 mm, confirmed this evidence. Furthermore, these results are in agreement with previous studies on population structure performed at Su Pallosu Bay (*Guala et al., 2006*) and Tavolara—Punta Coda Cavallo (*Guala, Simeone & Baroli, 2009*) (Fig. S6).

More interestingly, our results suggest the existence of a strong connection between the contribution to the reproductive potential of the fertile size classes and the pressure of commercial harvesting. Since the mean density of the US individuals at the HP zone was similar to the mean density of the CS sea urchins at the LP zone, and vice versa, the mean gamete output of the population (popMGO) was similar at the two zones but the contribution of the two fertile size classes was specular. The US individuals were therefore the main producers of gametes at the HP zone while the CS ones were likely annihilated by harvesting. On the other hand, at the LP zone, the main producer was the CS class probably because the US individuals were strongly reduced by fish predation as a response to protection measures (*Sala, 1997*; *Hereu et al., 2005*; *Parravicini et al., 2010*).

In contrast, the total gamete output of the population (popTGO), which represented the annual reproductive contribution of populations throughout the year, was two-fold higher at the LP zone. *Mita et al. (2007)* suggested that gonad size increases volumetrically with sea urchin test diameter, with the largest body size implying the most mature and developed gonads (and the consequent possibility to produce more than one cohort of gametes). Effectively, in marine reserves, more fertile gametes are produced by the largest sea urchins compared to sites where large sea urchins are lacking (*Lundquist, 2000*). Moreover, the very important effects related to the individual's age and size are proven by studies conducted

on different fish species that have demonstrated how a greater age diversity in a spawning stock may extend spawning both temporally and spatially. This, in consequence, increases the chances that more offspring encounter favourable conditions for development (*Secor, 2000*; *Berkeley et al., 2004b*; *Fiorentino et al., 2008*).

Despite the fact that a single year of study is not enough to unambiguously establish the regularity of reproductive episodes throughout time, the number of spawning events we observed at the two zones are consistent, respectively, with previous GSI studies conducted at Tavolara—Punta Coda Cavallo (*Guala, Simeone & Baroli, 2009*) and in a location in the Sinis peninsula, contiguous to Su Pallosu Bay (*Baroli et al., 2006*), that is intensely affected by fishing (*Marra et al., 2016*) (Fig. S7). Thus, since the popMGO strongly depended on the density per m$^2$ of fertile individuals regardless of their size class, it would be plausible that the continuous catch of commercial-sized sea urchins has led to a reduction of spawning events due to a considerable decrease of the body size of the fertile individuals. Hence, it might be reasonable to suppose that spawning events were favoured by the well-structured spawning stock and larger sea urchins which are typical of protected populations.

In general, sea urchin population dynamics is driven by various ecological processes that operate on different spatial and temporal scales. Larval supply fluctuates widely from region to region as it is associated with oceanic currents (*Fenaux, Cellario & Rassoulzadegan, 1988*; *Prado et al., 2012*). The success of settlements is influenced by local constraints linked to habitat (e.g., adult abundance, presence of crustose algae, substrate rugosity) (*Boudouresque & Verlaque, 2007*; *Oliva et al., 2016*). Finally, predation becomes the prevalent mechanism of sea urchin population control after settlement and serves as a critical bottleneck for urchin populations (*Guidetti, 2004*; *Hereu et al., 2005*; *Farina et al., 2009*; *Farina et al., 2014*; *Boada et al., 2015*). In our case, the lack of predatory fish in the HP zone (*Marra et al., 2016*; *Oliva et al., 2016*) and the high pressure of harvesting are likely to shift the whole potential reproductive contribution onto the young adults. On the contrary, in the LP zone, it mostly depended on commercial-sized individuals since predation in marine reserves significantly affects sea urchins until they reach the size of 50 mm TD (*Guidetti, 2004*).

Therefore, we could state that the self-sustenance of the sea urchin populations did not change between the two zones, despite the different harvesting pressure. The high number of recruits at the HP zone ensured an availability of new juveniles for the immediate future and suggested an important larval input of which the origin still has to be investigated. However, the two spawning events at the LP zone suggested a more abundant and successful gamete production than in the population which was under strong fishing pressure. This is probably due to a well-structured spawning stock which enhanced the resilience of the pristine population. Hence, in equal recruitment conditions, when the density of fertile individuals decreased (e.g., as a result of intense commercial harvesting or natural predation) the mean reproductive contribution consequently diminished.

Because of over-exploitation of *P. lividus* populations in several Mediterranean areas (*Pais et al., 2012*; *Bertocci et al., 2014*) and their ecological (*Sala, Boudouresque & Harmelin-Vivien, 1998*) and economic relevance (*Palacín et al., 1998*), sea urchin fisheries need urgent and effective regulation. Our findings may have important implications for creating a

management of sea urchin fisheries that is effective in terms of the sustainable conservation of resources.

In a marine reserve context, where sea urchin populations have a normal bimodal structure (*Brundu et al., 2013*), the opening of commercial harvesting should be avoided (despite the insistence of local fishermen) to prevent the depletion of fertile individuals, as a result of the synergistic action with the natural predation, and the breakdown of the population. On the contrary, in areas affected by fishing pressure, larger individuals are lacking (*Baroli et al., 2006*), but the survival of the fertile, intermediate-size classes, which therefore support the mean gamete output of the whole population, is guaranteed by laws (as harvesting is authorized only for individuals $\geq 50$ mm) and by the absence of predatory fishes, which in turn are targeted by commercial fishing.

Our present findings suggest that the reproductive potential contribution of the population does not depend on the size of fertile individuals but on their density. For this reason, the harvesting of individuals between 40 mm and 50 mm should be avoided particularly in over-fished locations, since they are the only ones capable of generating new life, as far as we know without more information on the origin of the larvae present.

## ACKNOWLEDGEMENTS

We are grateful to the staff of Tavolara—Punta Coda Cavallo Marine Protected Area and to B Cristo for assistance and help in the field. MA Figus, A Pinna, D Vallainc, ML Vitelletti are gratefully acknowledged for helping in lab activities. We would like to thank L Fazioli for providing us sea surface temperatures. Discussions with G Ceccherelli and S Guerzoni, during the various stages of this study, were of great help and much appreciated. Furthermore, the authors would like to acknowledge the contribution of the anonymous reviewers who significantly helped to improve the original manuscript.

### Funding

This research was funded by a Regione Autonoma Sardegna (L/7 CRP49692) grant. The funders had no role in study design, data collection and analysis, decision to publish, or preparation of the manuscript.

### Grant Disclosures

The following grant information was disclosed by the authors:
Regione Autonoma Sardegna: L/7 CRP49692.

### Competing Interests

The authors declare there are no competing interests.

### Author Contributions

- Barbara Loi and Simone Farina performed the experiments, analyzed the data, wrote the paper, prepared figures and/or tables, reviewed drafts of the paper.

- Ivan Guala conceived and designed the experiments, performed the experiments, analyzed the data, wrote the paper, prepared figures and/or tables, reviewed drafts of the paper.
- Rodrigo Pires da Silva performed the experiments, analyzed the data.
- Gianni Brundu performed the experiments, reviewed drafts of the paper.
- Maura Baroli conceived and designed the experiments, contributed reagents/materials/-analysis tools, reviewed drafts of the paper.

## Field Study Permissions

The following information was supplied relating to field study approvals (i.e., approving body and any reference numbers):

Field samplings were approved by the Regione Autonoma Sardegna (fishing license for scientific purposes n. 9727/AP SCIE/N.7 03/06/2016).

## Data Availability

The raw data has been supplied as a Supplementary File.

## Supplemental Information

Supplemental information for this article can be found online at http://dx.doi.org/10.7717/peerj.3067#supplemental-information.

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
