# Peer review of "Hard time to be parents? Sea urchin fishery shifts potential reproductive contribution of population onto the shoulders of the young adults"

_PeerJ, doi:10.7717/peerj.3067_

## Round 0.1 · original submission · Major Revisions

Two reviewers have provided constructive comments on your manuscript. While both were positive, one mentioned additional data that should be supplied (sex ratio, GSI, etc), while the other offered excellent statistical analyses advice. Based on these comments, my decision is 'major revisions' are needed.

Reviewer 1 ·

Basic reporting

No comment

Experimental design

Authors indicated that no sampling was done in November, because of the weather condition. However, it is important to have data for this month, because the fishermen are collecting sea urchins starting from November (starting month is very important) as authors said in line 119
"practiced from 119 November to April by 189 professional fishermen authorized"

Validity of the findings

This article is lacking a very important information/calculations for instance:

1- The title is "Hard time to be parents? Sea urchin fishery shifts potential reproductive contribution of population onto the shoulders of the youngest", therefore you should calculate the size at first maturity to detect condition of the stock in the areas of study and to estimate adult and young individuals within population.

3- GSI estimated in the present study, but sex ratio are important too.

2- The authors indicated that the take area under fishing pressure, therefore the calculations of catch per unit effort and exploitation rates are mandatory. In addition, some calculations are important too...... such as growth rate and fecundity.

Additional comments

See attached!

Reviewer 2 ·

Basic reporting

The authors should edit the manuscript, particularly the introduction, with an eye for proper paragraph structure and topic sentences.

51-77: There’s more than 1 paragraph here. Authors should consult a text on scientific writing and focus on using only one topic sentence per paragraph

Experimental design

The experimental design is fine as long as the authors are not testing for differences between protected areas and harvested areas. Any analysis with Protection as a factor is flawed.

The authors have samples from two areas that happen to have different status for harvesting. There is subsampling of these two locations, but there is no replication of closed and open areas.

There is site a and site b. It just so happens That a is protected and b is not. But there needs to be site c and d if you want to test protection level.

Table 1. There is no replication for protection level, thus it cannot be a factor in the ANOVA. This is a classic case of pseudo-replication if the authors are trying to say that urchins in protected areas are different than urchins in harvested areas.. I don’t think this is a fatal flaw for the manuscript though, because the results don’t hinge upon proving a difference between protected and unprotected areas. If Protection is changed to Location or Region then it should be ok

Validity of the findings

The findings are valid.

---

## Round 0.2 · accepted · Accept

I have gone over the manuscript myself, and have found no outstanding issues - it was well revised. I look forward to seeing this work published!

Reviewer 1 ·

Basic reporting

Please, check the English language/structure again.

Experimental design

no comment

Validity of the findings

no comment

Reviewer 2 ·

Basic reporting

Acceptable

Experimental design

Adequate and appropriate

Validity of the findings

Reasonable, valid